# Dense CTD survey versus glider fleet sampling: comparing data assimilation performance in a regional ocean model West of Sardinia

Jaime Hernandez-Lasheras[1] and Baptiste Mourre[1]

[1]Balearic Islands Coastal Observing and Forecasting System - SOCIB

*Correspondence to:* Jaime Hernandez Lasheras (jhernandez@socib.es)

**Abstract.**

The REP14-MED sea trial carried out off the West coast of Sardinia in June 2014 provided a rich set of observations from both ship-based CTDs and a fleet of underwater gliders. We present the results of several simulations assimilating data either from CTDs or from different subsets of glider data, including up to eight vehicles, in addition to satellite sea level anomalies, surface temperature and Argo profiles. The WMOP regional ocean model is used with a Local Mutimodel Ensemble Optimal Interpolation scheme to recursively ingest both lower-resolution large scale and dense local observations over the whole sea trial duration. Results show the capacity of the system to ingest both type of data, leading to improvements in the representation of all assimilated variables. These improvements persist during the three day periods separating two analyses. At the same time, the system presents some limitations in properly representing the smaller scale structures, which are smoothed out by the model error covariances provided by the ensemble. An evaluation of the forecasts using independent measurements from shipborne CTDs and a towed Scanfish deployed at the end of the sea trial shows that the simulations assimilating initial CTD data reduce the error by 39% on average with respect to the simulation without data assimilation. In the glider-data-assimilative experiments, the forecast error is reduced as the number of vehicles increases. The simulation assimilating CTDs outperforms the simulations assimilating data from one to four gliders. A fleet of eight gliders provides a similar performance as the 10-km spaced CTD initilization survey in these experiments, with an overall 40% model error reduction capacity with respect to the simulation without data assimilation when comparing against independent campaign observations.

## 1 Introduction

Short-term regional ocean prediction is important to respond to maritime emergencies related to search-and-rescue or accidental contamination, for maritime security or as a support to naval operations. High-resolution regional ocean circulation models are used to downscale the conditions provided by operational large scale models, so as to represent mesoscale and coastal processes which are not properly resolved in the large scale simulations but play a major role in ocean transports of relevance for practical applications. Data assimilation (DA), which aims at optimally combining dynamical ocean models with in-situ and remotely sensed observations, constitutes an essential component of the prediction systems since it helps to recursively improve the initial conditions used for the prediction phases.

In order to constrain errors and remain as close as possible to reality, models must be fed with different kinds of observations. Satellites play a key role, providing regular near real time data of surface variables such as temperature and sea surface height. Water column measurements are more scarce. The Argo program provides routine temperature and salinity profiles at regular intervals, but the distance between floats is insufficient to monitor the mesoscale and finer scale variability (Sánchez-Román et al., 2017). Dedicated campaigns providing underwater measurements from ship data or glider measurements provide complementary data over specific areas. Efficient DA systems should be able to advantageously combine large scale observations over a large domain with more dense, high-resolution observations in specific areas. Traditionally, campaigns onboard research vessels (RV) have been carried out to collect dense CTD data to initialize regional ocean prediction systems. However, campaigns are not always possible. They depend on ship availability, weather, access to the area of interest and they remain very expensive. Recent evolutions in technology allow to deploy autonomous underwater vehicles (AUV) such as gliders to collect dense hydrographic data over specific areas of interest (Testor et al., 2010; Ruiz et al., 2012; Rudnick, 2016; Liblik et al., 2016). Gliders are able to operate under hard maritime situations and to reach difficult access areas, with an overall cost reduced compared to traditional ship campaigns. Glider missions are typically planned to reach a series of locations commonly called waypoints, in order to track areas of interest and adapt to safety conditions (Garau et al., 2009). Their controllability also permits adaptive sampling procedures, changing their route along the mission with the objective to collect data at optimal locations to maximize their information content (e.g. Lermusiaux (2007); Mourre and Alvarez (2012)).

The potential of gliders to sample fine-scale processes and to identify different water masses has been demonstrated (Pascual et al., 2017; Heslop et al., 2012), as well as their capability to improve ocean model predictions via DA (e.g. Melet et al. (2012); Shulman et al. (2008); Pan et al. (2014); Mourre and Chiggiato (2014)). The question arises whether the sampling offered by a fleet of several gliders is as useful as a traditional ship-based CTD survey for regional ocean forecasting applications. This is the question we are addressing in this paper.

The rich dataset collected during the REP14-MED campaign is used for this purpose. REP14-MED took place in June 2014 offshore the western coast of Sardinia (Onken et al., 2018; Knoll et al., 2017). Two RV tracked in parallel a 100 x 100 km$^2$ area during a 20-day period, providing dense CTD sampling with a 10-km separation and continuous towed CTD measurements for limited periods of time. In addition, a fleet of 11 gliders were deployed performing back-and forth sections perpendicular to the coast with a 10 km vehicle intertrack distance.

The Sardinian Sea is a region of the so-called Algero-Provençal basin of the Western Mediterranean Sea. In this region, the surface layer is characterized by a water mass of Atlantic origin and a strong mesoscale activity. The region is one of the most dynamic areas of the entire Mediterranean Sea (Olita et al., 2011; Millot, 1999), since it is strongly influenced by instabilities of the Algerian current. This generate intense anticyclonic eddies which can propagate northward towards the western Sardinian coast (Robinson et al., 2001; Testor et al., 2005; Escudier et al.). Such eddies can last from weeks to months. They are responsible for an intense mesoscale activity in the study region (Santinelli et al., 2008). A southward current flowing along the southern part of the Sardinian coast has also been evidenced in long-term numerical studies (Olita et al., 2013) and in field campaigns contributing to episodically wind-induced advection of coastal water (Ribotti et al., 2004). At depth, eddies

are also generated from the interaction between the Algerian Gyre and inflows of Levantine Intermediate Water (LIW) and Tyrrhenian Deep Water (TDW) coming from the Sardinia Channel (Testor et al., 2005).

The assimilation system used in this work follows an EnOI (Ensemble Optimal Interpolation) scheme (Evensen, 2003). This method provides a cost-effective approach when compared with more advanced methods such as EnKF (Ensemble Kalman Filter) or 4Dvar (Oke et al., 2008), which is suitable for operational implementations in regional ocean models. EnOI is a 3-dimensional sequential DA method. A stationary ensemble of model simulations is used to calculate background covariances. Contrary to the EnKF which requires to evolve an ensemble of simulations, a single model integration is only necessary between two analyses steps in the EnOI, making the method numerically efficient. The EnOI provides a suboptimal solution compared to the EnKF (Sakov and Sandery, 2015). However, it represents a good alternative allowing to use a large ensemble size together with localization when necessary (Oke et al., 2007). In this work, a Local Multimodel EnOI scheme is implemented. "Multimodel" represents the fact that the library of ocean states is built using different long-term model simulations. "Local" means that the EnOI analysis comprises some domain localization to reduce the impact of potential significant covariances associated with remote observations.

The paper is organized as follows: Section 2 presents the observing and modelling frameworks, as well as the specific forecast experiment. Section 3 details the results, which are further discussed in Section 4. Finally, Section 5 concludes the paper.

## 2 Data and Methods

For this study, several simulations were produced assimilating different datasets from the REP14-MED campaign. This section describes the model and data used and the methodology followed in this work.

### 2.1 REP14-MED experiment

The REP14-MED sea trial (Onken et al., 2018) was conducted in the framework of the EKOE (*Environmental Knowledge and Operational Effectiveness*) research program of the Centre for Maritime Research and Experimentation (CMRE, Science and Technology Organization - NATO). It is part of a series of sea trials dedicated to *Rapid Environmental Assessment* (REA), denoted by the acronym REP (*Recognized Environmental Picture*). Led by CMRE and supported by 20 partners, the trial took place during 20 days in June 2014, with RV *Alliance* and *Planet* conducting a joint survey over an approximately $100 \times 100$ km$^2$ area off the west coast of Sardinia (Figure 1). A massive amount of data was collected during the campaign with various oceanographic instruments, including CTD stations, towed Scanfish and CTD chain, ship mounted ADCP, shallow and deep underwater gliders, moorings, surface drifters and profiling floats. The sampling was divided into three legs. The time distribution of the collection of observations used in the present work is illustrated in Figure 2.

During Leg 1, both RV conducted a parallel sampling of the target area, collecting CTD data with a 10 km distance between stations over a five day period. During Leg 3, CTD data were collected with the same density, yet over a reduced spatial extension, providing very valuable data to validate the forecast experiments. CTD casts reached down to 1000 m deep when

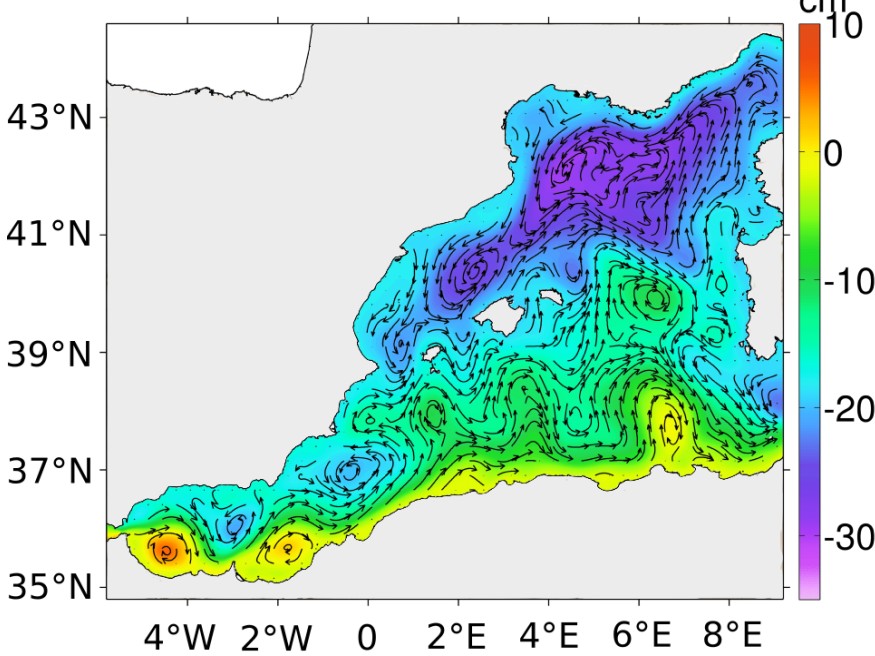

**Figure 1.** Sea surface height annual mean for year 2014 and corresponding surface geostrophic currents from WMOP model. The REP14-MED sea trial area is highlighted in red.

possible. A few profiles even get deeper in order to characterize deep water masses. Additional towed Scanfish measurements of temperature and salinity down to 200 m depth allowed to complete the characterization of the area during Leg 3. At the same time, and during the whole duration of the campaign, eight gliders were considered, travelling continuously along back and forth transects perpendicularly to the Sardinian coast. Five of these gliders were deep gliders submerging to depths down to 5   800 m, the remaining three were shallow water platforms collecting data in the upper 200 m only. Each of these single transects was completed in about three days for each way. Notice that three additional gliders were deployed during the sea trial, but due to technical problems, duplication of the track and lack of processed data, they were discarded here. All glider tracks are approximately parallel to each other, with an intertrack distance around 10 km, thus covering the target area. Figure 2 shows the position of CTD, glider and Scanfish data during Legs 1 and 3 of the sea trial.

10  **2.2   Model**

The model used in this work is the Western Mediterranean OPerational model (WMOP, Juza et al. (2016)), covering a domain extending from Strait of Gibraltar to the Sardinia channel. WMOP is based on ROMS (Shchepetkin and McWilliams, 2005), a three-dimensional free-surface, sigma coordinate, primitive equations model using split-explicit time stepping with Boussinesq and hydrostatic approximations. WMOP is set-up with 32 vertical levels and a 2 km spatial resolution. It is forced 15  at the surface using the 3-hourly and 5-km resolution HIRLAM atmospheric model provided by the Spanish meteorological

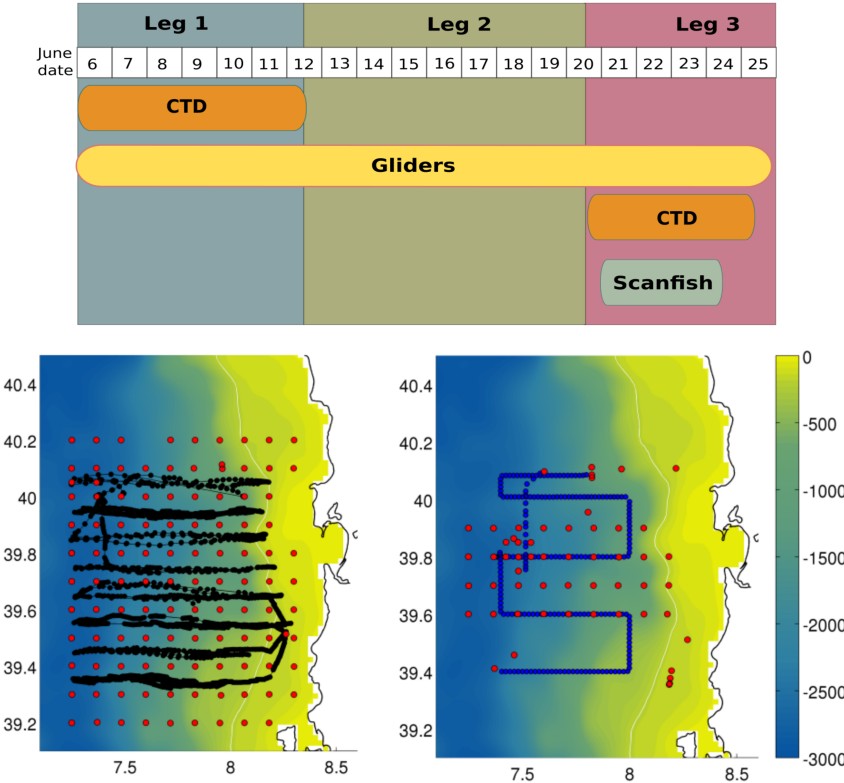

**Figure 2.** Top panel: Sampling schedule in June 2014 of the REP14-MED sea-trial data employed in the present work. The spatial distribution of observations is illustrated in the two bottom panels. CTDs from Leg 1 (red) and gliders (black) observations used for assimilation are shown in the bottom left panel. Independent CTDs (red) and Scanfish (blue) gathered during Leg 3 and used for the validation are shown in the bottom right pannel. The colorbar indicates depth (m) and the white contour indicates the 200m isobath.

agency (AEMET). In this work, the initial state of the forecast experiments is provided by the simulation fields on 1 June 2014 of a seven-year long free run WMOP simulation spanning the period 2009-2015. This simulation uses initial state and boundary conditions from the CMEMS-MED reanalysis (Simoncelli, 2014). In addition, several WMOP free run hindcast simulations were generated including modifications of the parent model used as initial and boundary conditions and some model parameters. These different simulations provide the library of ocean states used by the DA system.

The model has been evaluated using satellite and in-situ observations (Mourre et al. (2018), in press). The mean circulation of the free run simulation over the year 2014 is illustrated in Figure1). It is found to properly represent the mean surface geostrophic circulation over the basin, in particular the main features which are the Alboran Gyres, the Algerian Current along the African coast and its associated instabilities, the Northern Current along the French and Spanish coast and the Balearic Current flowing north-eastwards north of the Balearic Islands. Close to Sardinia, the mean circulation in the model

is characterized by a south-eastward flow centered around 40N, which separates into two branches flowing northward and southward, respectively, when approaching the Sardinian coast, also giving rise to small eddies in the REP14-MED area. This is in agreement with both the historically established regional surface circulation (Millot, 1999) and the more recent average estimates provided by the mean dynamic topography (Rio et al., 2014).

5        Moreover, we illustrate here the sea surface temperature (SST) maps derived from the model and satellite data at the beginning of the REP14 period (Figure 3). The model is found to properly represent the large sale spatial variability of the SST, with lower temperatures around the Gulf of Lions, warmer waters in the Algerian basin south of the Balearic Islands and some lower temperature inflow of Atlantic Water from the Strait of Gibraltar. Finer details associated with mesoscale eddies and filaments do not generally coincide between the free run model and observations. In particular local differences are found in the
10    REP14-MED area, with the model slightly overestimating surface temperatures due to an apparent more pronounced advection of higher temperature waters from the south-west.

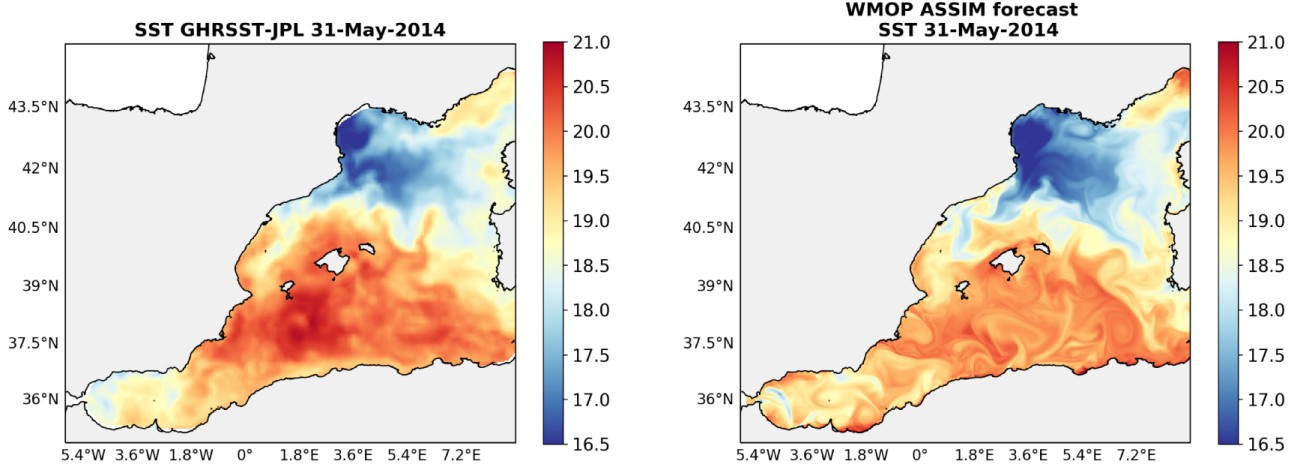

**Figure 3.** SST for 31 May 2014 from 1) left: GHRSST JPL MUR satellite-derived product, 2) free run WMOP model .

## 2.3    Data Assimilation System

The WMOP DA system is based on a Local Multimodel Ensemble Optimal Interpolation (EnOI) scheme. It consists in a sequence of analyses (model updates given a set of observations) and model forward simulations. During the analysis step,
15    the state vector $\mathbf{x}^a$ is updated according to eq.(1), where $\mathbf{x}^f$ is the background model state vector, $\mathbf{H}$ is the linear observation operator projecting the model state onto the observation space and $\tilde{\mathbf{K}}$ is the Kalman gain estimated from the sample covariances

(eq.2). $\mathbf{y}$ is the vector of observations. Matrices $\tilde{\mathbf{P}}^f$ and $\mathbf{R}$ are the error covariance matrices of the model and the observations, respectively.

$$\mathbf{x}^a = \mathbf{x}^f + \tilde{\mathbf{K}}(\mathbf{y} - \mathbf{H}\mathbf{x}^f) \tag{1}$$

$$\tilde{\mathbf{K}} = \tilde{\mathbf{P}}^f \mathbf{H}^T (\mathbf{H}\tilde{\mathbf{P}}^f \mathbf{H}^T + \mathbf{R})^{-1} \tag{2}$$

$\tilde{\mathbf{P}}^f$ contains the background error covariances (BECs) estimated by sampling three long-run simulations of the WMOP system with different initial/boundary forcing (coming from CMEMS MED and GLOBAL models) and momentum diffusion parameters. More concretely, for each analysis, a 80-realization ensemble is generated to calculate the BECs. Ensemble realizations are multivariate model fields sampled from the three simulations during the same season, with a time window of 90 days centered on the analysis date. The seasonal cycle is removed from the ensemble anomalies to discard the corresponding large

scale correlations mainly affecting temperature. Following this procedure, the computed BECs reflect the spatial variability and anisotropy of the ocean mesoscale circulation. They also represent dynamically consistent covariances between different model variables and depths. Moreover, a domain localization strategy (Ott et al., 2004) is used to dampen the impact of remote observations. A 200 km localization radius is used, determined by both the size of mesoscale structures and the approximate distance between two Argo platforms in the Western Mediterranean basin. Here, the domain localization consists in computing

independent analyses for each water column of the WMOP domain, considering only the observations located within a 200 km radius. It allows to locally dampen the impact of remote observations in the presence of spurious long-range correlations. The code used in this study is written in C and is an adaptation of the EnKF version used in Mourre et al. (2006b), Mourre et al. (2006a) and Mourre and Chiggiato (2014). It was also previously used during the Alborex experiment carried out in the Alboran Sea (Pascual et al., 2017).

In the EnOI, as in any other sequential DA scheme, a special care needs to be brought to the model initialization after analysis updates (Oke et al., 2008). When restarting the simulation from an analysis field, the multivariate initial fields may be violating some physical constrains, such as mass conservation. The model response to balance this state may generate some spurious waves and introduce noise into the system. To minimize such effects, a nudging strategy has been implemented to restart the model after the analysis. In concrete terms, after an analysis is computed, the model is restarted 24 hours before the

analysis date applying a strong nudging term in the temperature, salinity and sea level equations towards analyzed values. The time scale associated with this nudging term is one day. The nudging procedure reduces the model correction, but guarantees updated multivariate fields closer to the model equation balances, which limits instabilities.

  The assimilation system implemented here uses a three-day cycle (Figure 4), which was determined according to the time spent by the gliders to complete one zonal transect. All measurements collected during these three days are considered synoptic

in the DA process. Altimeter sea level anomalies (SLA), SST and Argo temperature and salinity profiles are assimilated over the whole WMOP domain. For each analysis, a five day time window in the past is defined to select Argo observations. This

window corresponds to the interval between two profiles provided by a single platform. This ensures that every model point is bounded by at least one Argo profile within the 200 km localization radius during each analysis. Concerning altimetry, the last 72-hour CMEMS along-track reprocessed filtered sea level observations are considered for the analysis. The SST field is given by the daily L4 near real time GHRSST JPL-MUR satellite-derived interpolated product (https://podaac.jpl.nasa.gov/dataset/

JPL-L4UHfnd-GLOB-MUR, last access 28 August 2018). The last available field before analysis is considered. The original 1 km resolution data is smoothed and interpolated onto a 10 km resolution grid to limit the number of observations considered for each analysis. The selected resolution is considered to be sufficient to represent the main circulation features and mesoscale structures present in this SST product, permitting at the same time an affordable computational cost.

The glider profiles are considered as vertical. The corresponding observations are binned vertically and a single value is given

for each model grid cell. The representation error is the addition of a vertical and a horizontal components. For each vertical level, the observed variance in the vertical grid cell is used as an approximation of the vertical representation error. In addition, the horizontal representation error variance is assumed to be 0.0625 $K^2$ and 0.0025 for temperature and salinity measurements, respectively. CTD observations are binned vertically in a similar way before assimilation, considering the representation error in an analogous way.

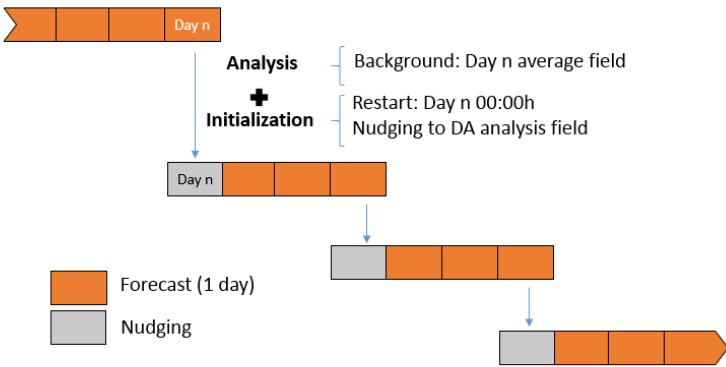

**Figure 4.** Scheme of the three day DA cycles.

Figure 5 illustrates the innovations (differences between the observations and the background model) computed for the first analysis carried out on 31 May. It shows the absence of significant biases in the model prior to DA, which is a prerequisite for an effective assimilation of the observations. Moreover, the magnitude of the standard deviation of innovations of the surface variables (0.42 K for SST and 0.042 m for SLA) properly matches that of the observation error (0.56 K for SST and 0.036 m for SLA) and the ensemble spread (1.10 K for SST and 0.056 m for SLA for the analysis on 31 May), which guarantees the

necessary overlap between the probability density functions of model and data.

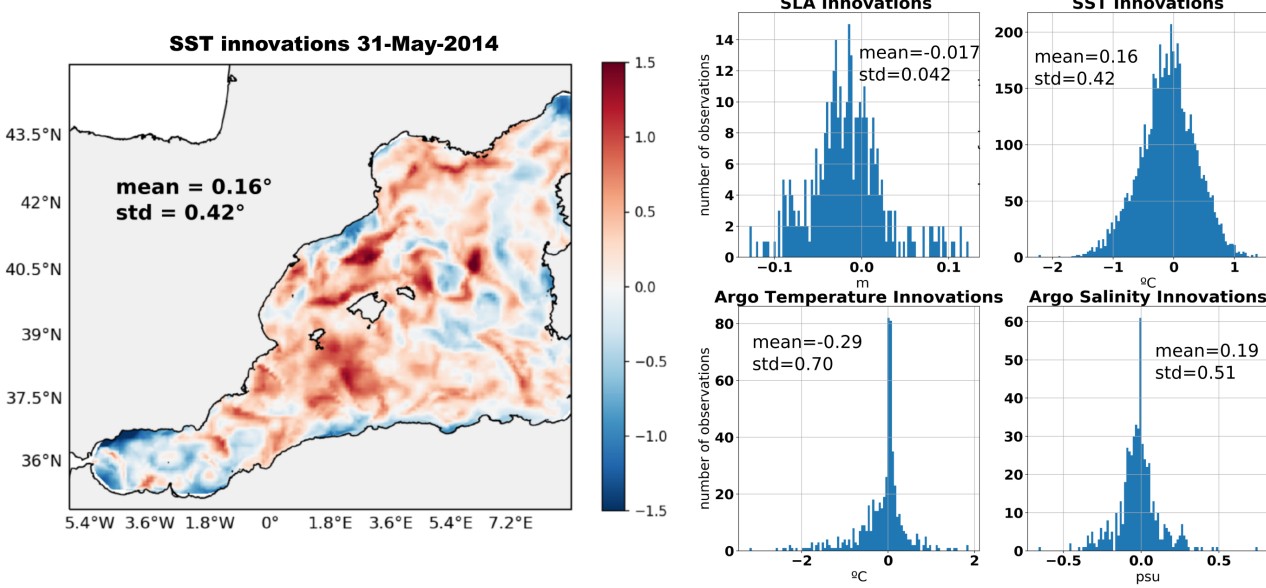

**Figure 5.** Left: SST misfits between the observations and the free run model on 31 May 2014. Right: histograms of the innovations for the different sources of observation ingested by the assimilation system. The corresponding mean and standard deviation are provided in each panel.

## 2.4 Experiments

In addition to the background simulation without any DA (hereafter NO_ASSIM), seven simulations were produced spanning the period 1-24 June 2014, assimilating different sets of observations. The first simulation (GNR) assimilated generic observations from satellite along-track SLA, satellite SST and Argo temperature and salinity profiles. The second simulation
5   (GNR_CTD) assimilated these generic observations plus all CTD temperature and salinity profiles collected during Leg 1. The five remaining simulations assimilated the generic observations plus glider temperature and salinity data from one to eight vehicles (GNR_1G, GNR_2G, GNR_3G, GNR_4G, GNR_8G), selected among the available platforms to optimally cover the area of interest. For example, GNR_1G considers the glider which travels in the center part of the domain, GNR_2G selects the two gliders which divide the study region in three areas of similar dimensions, etc... The different sets of vehicles selected
10  for these simulations are illustrated in Figure 6.

The whole timeline of the numerical experiments is described in Figure 7. A spinup period of nine days was imposed for all these data-assimilative simulations, during which only the generic observations were assimilated. As explained previously, the three-day assimilation cycle implemented in this study corresponds to the time spent by a glider to complete a zonal transect. In the case of the CTDs, as the duration of the data collection in Leg 1 was six days, the data was assimilated during two cycles.

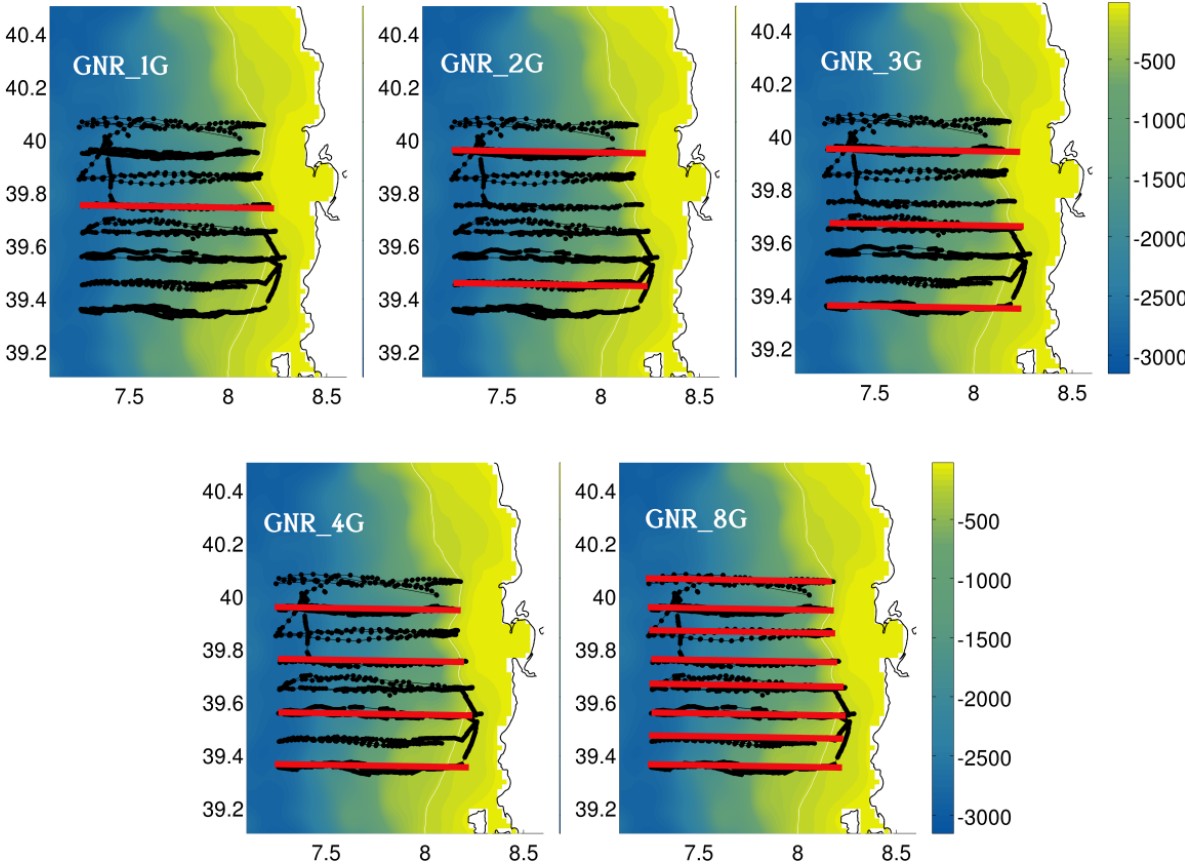

**Figure 6.** Illustration of the different sets of gliders selected in the different data-assimilative experiments. The position of all glider measurements are shown in black dots. The red zonal lines indicate the selected glider tracks in each of the experiments. The name of the corresponding simulation is specified in each panel.

After the last analysis, both Leg 3 CTDs and Scanfish temperature and salinity measurements are used as independent observations to evaluate the performance of the simulations.

## 3 Results

We evaluate in this section the performance of the DA following three successive steps. We first verify that the data from the different sources are properly ingested in the system over the whole modelling area both during the spinup period and subsequent assimilation phase. Then, we examine the impact of the assimilation of the local and dense observations datasets

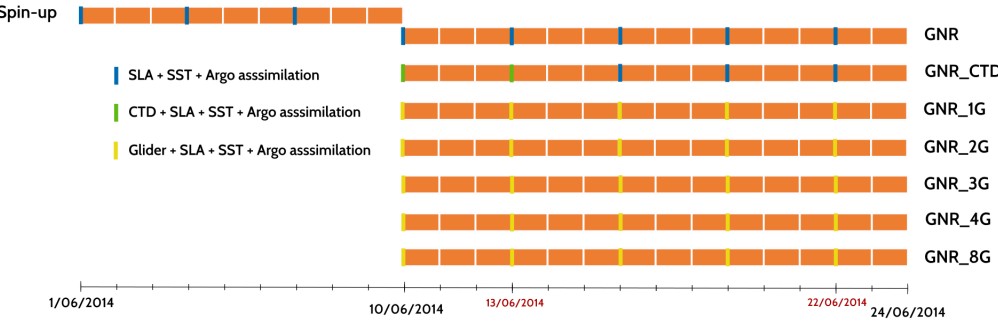

**Figure 7.** Timeline of the seven data-assimilative simulations. The analyses dates are highlighted in color, indicating the assimilated datasets.

onto the temperature, salinity and density fields in the REP14-MED area. Finally, we assess the performance of the simulations against independent data from CTDs and Scanfish observations collected during Leg 3.

### 3.1 Data ingestion and performance over the whole modelling area

We first assess here the performance of the assimilation during the spinup period by analyzing the evolution of the Root-Mean-
Square-Difference (RMSD) between the model simulations and satellite SLA, SST observations and Argo profiles. For each source and variable, the RMSD is calculated as expressed in eq.(3) below, where $o_i$ and $m_i$ take the values of the observations and their model equivalents, respectively. $n$ is the number of observations. To better highlight relative simulation improvements, the RMSD for each specific day is normalized by dividing the RMSD by that of the simulation without any DA for that specific day. A reduction of the normalized RMSD indicates that the analyzed field is closer to the observations than the background
field without assimilation.

$$RMSD = \sqrt{\frac{\sum_{i=1}^{n}(o_i - m_i)^2}{n}} \qquad RMSD_{normalized} = \frac{RMSD_{assim\_simulation}}{RMSD_{no\_assim}} \qquad (3)$$

The normalized RMSD is computed every day from 1 to 9 June. For the days including an analysis, the observations assimilated during this analysis are used to compute the normalized RMSD. This includes the different assimilation windows (five days for Argo, three days for SLA, one day for SST in particular). For the remaining days, we consider the observations
that the system would have ingested if we had performed the analysis on that date, so considering similar time windows. Model equivalents to the observations are obtained through linear interpolation in space of the average daily model fields onto the position of the measurements.

The results are presented in Figure 8. They show a satisfactory and continuous reduction of the RMSD for all the sources of data and variables, indicating a good system performance. The normalized RMSD is significantly reduced during the first
analysis (between 20 and 60% depending on the analyzed variable), it then tends to slightly increase until the next analysis

three days later, which reduces it again in most of the cases. In some occurrences, the RMSD continues decreasing during two days after the analysis. The overall persistence of the correction between two assimilation dates is especially remarkable. It reveals the general proper performance of the assimilation system, which is able to recursively correct the multivariate fields without introducing spurious structures and instabilities which would significantly alter the system.

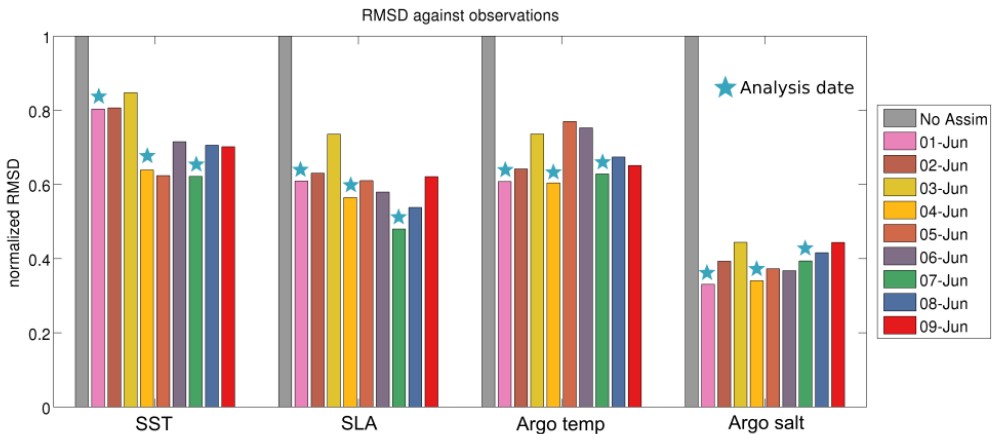

**Figure 8.** Evolution of the normalized RMSD against observations for the spinup simulation.

After this verification of the overall satisfactory performance of the DA during the spinup period in terms of RMSD, the same kind of assessment was performed for the seven subsequent simulations, the GNR control simualtion and the ones assimilating either CTDs or glider observations during the field experiment besides the generic observations. We only show here results from the GNR_CTD simulation (Figure 9), since the behaviour is very similar for the rest of the simulations. The system still properly reduces the normalized RMSD in terms of SST, SLA and T-S profiles at Argo locations, with similar reductions as that observed
during the spinup period (from 20 to 60% of error reduction). An important aspect in this comparison is that the assimilation of high-resolution T-S profiles data in the REP14-MED area does not negatively affect the overall performance of the system over the whole modelling area. This could happen through the generation of spurious structures in the densely observed area which could then propagate over the domain. Moreover, the reduction of the normalized RMSD with respect to CTD observations shows that the local observations have also been properly ingested in the system. Notice that the relatively larger SLA RMSD
found during the period 10-23 June compared to the spinup period also affects the GNR simulation. Therefore, it is not due to the incorporation of CTD observations, but rather related to the natural evolution of SLA errors.

### 3.2    Temperature, salinity and density fields in the REP14-MED area

To complement these statistical diagnostics based on the normalized RMSD, we analyze here the temperature and salinity fields in the REP14-MED trial area on 13 June. This corresponds to the first day after the second analysis of CTD and glider
DA cycles. At that time, either all CTD data from Leg 1, or one back-and-forth transect from the gliders have been introduced into the system. Figure 10 shows temperature, salinity and potential density daily average fields at 50 m depth for four of the

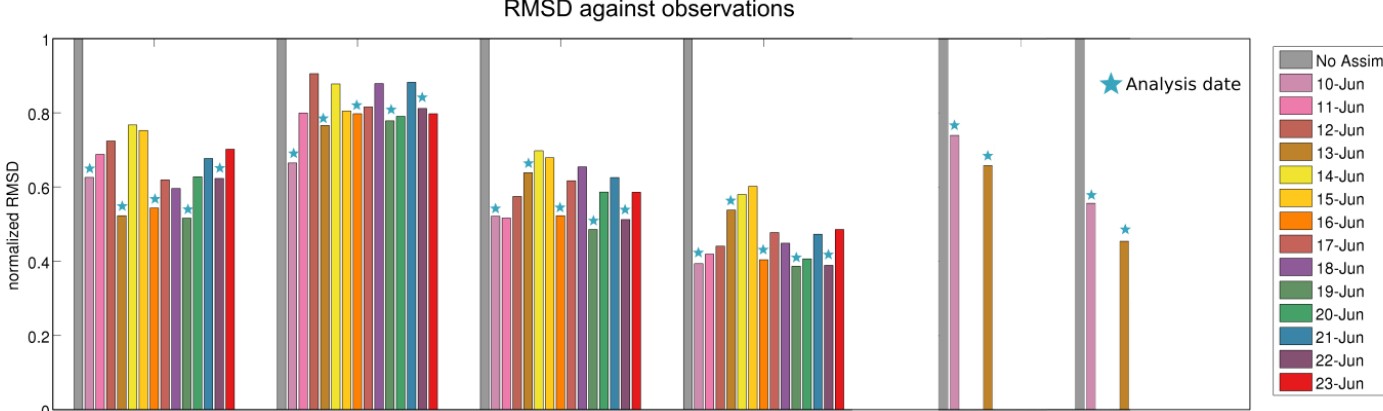

**Figure 9.** Evolution of the normalized RMSD against observations for the GNR_CTD simulation.

simulations (NO_ASSIM, GNR, GNR_CTD and GNR_8G) on 13 June. The temperature and salinity data assimilated until that date are also represented as colored dots on the panels corresponding to GNR_CTD and GNR_8G.

As illustrated by the potential density maps, two different water masses are represented in the NO_ASSIM simulation. While the northern part of the domain is mostly occupied by a denser water mass with a salinity over 38, lower density waters characterized by their relative fresher and higher temperature characteristics are found in the southern part of the REP14-MED domain. The GNR simulation redistributes these water masses over the domain, representing patches of denser water in the central, south-western, north-western and northern coastal parts of the domain. These patches, characterized by a higher temperature and salinity, are associated with cyclonic circulations.

The additional assimilation of dense local data from CTDs and gliders further modulates these patterns, producing smaller scale patches and filaments. Two main high temperature anomalies are detected in both CTD and glider observations at 50 m depth, associated with relatively small salty anomalies. The strongest one, located around 7.7°E-39.3°N, is somehow represented in both simulations GNR_CTD and GNR_8G, with a more pronounced signature in GNR_8G. Notice that this signature is not fully coincident with the observed location displayed in Figure 10 due to the evolution of the model from the first analysis on 10 June to the time of the plot three days later. The second relatively high temperature patch found around 7.5°E-40°N is less marked that the first one. Again, it is somehow better reproduced in GNR_8G than GNR_CTD. The observations, from both CTDs and gliders, are characterized by an energetic small scale variability, which translates into small scales and filamental structures in the model after DA. Notice also the improvements in the relatively higher salinity along the coast after assimilation of the measurements from the CTDs and the gliders. The relatively high salinity patch around 39.5°N seen in the GNR simualtion is strongly attenuated in both GNR_CTD and GNR_8G. The density fields of GNR_8G exhibit two areas of lower potential density associated with these two anomalies, in good qualitative agrement with the observations even if the magnitude of the gradients is reduced compared to the measurements. These density anomalies are less clear in GNR_CTD.

Both simulations show denser water on the north-eastern part of the domain and similar overall circulation patterns which significantly differ from NO_ASSIM and have also marked differences with GNR. A common property observed in both simulations is the current flowing northeastwards in the central part of the sampled area and bifurcating near the coast, with one branch directed southwards and other northwards, giving also rise to cyclonic and anticyclonic eddies with dimensions around 30-40 km.

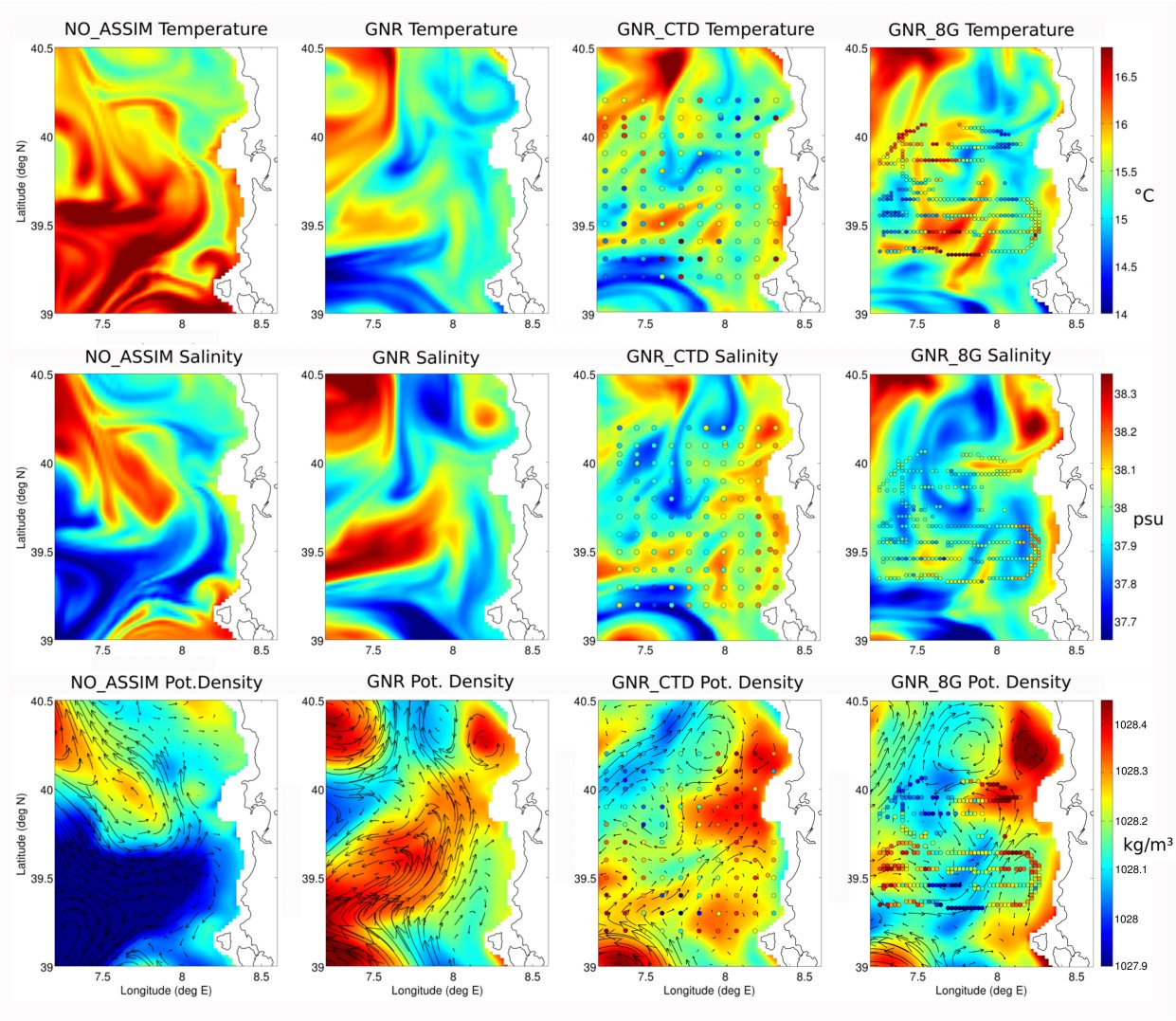

**Figure 10.** Model temperature (upper panels), salinity (middle panels) and potential density and currents (lower panels) at 50 m depth on 13 June. From left to right: simulations NO_ASSIM, GNR, GNR_CTD and GNR_8G. The assimilated data are superimposed as colored dots in the temperature and salinity panels for the two simulations GNR_CTD and GNR_8G.

### 3.3 Performance assessment using independent data during Leg 3

As a third step we analyze here the realism of the simulations during Leg 3 using independent observations which have not been assimilated in the experiments. More specifically, we compared the model outputs on 22 June (after all assimilation cycles have been completed) with CTD and Scanfish temperature and salinity observations collected between 20 and 23 June. A qualitative
analysis is first performed, based on the potential density fields reconstructed from both CTD and Scanfish observations at 50 m depth. The DIVA software (Data Interpolating Variational analysis, (Barth et al., 2010)) and its web interface (http://ec.oceanbrowser.net/emodnet/diva.html , last access 28 August 2018) have been used to generate the interpolated density field. Figure 11 compares the density fields at 50 meter depth from the different simulations with this density field derived from the observations. Model currents at that depth are shown as well as the CTD and Scanfish observations.

The main features represented in the reconstructed density field derived from the observations include a marked negative density anomaly centered around 7.8°E-39.4°N with a spatial extension around 40 km, a coastal fringe with relatively denser waters, and a second patch of denser water between 39.5°N and 40°N on the Western side of the domain. These features were somehow already present during Leg 1 (see Figure 10 and Section 3.2).

All the data-assimilative simulations represent the denser coastal fringe and the associated southward flow, yet with different
characteristics. It extends offshore, associated with a cyclonic eddy, in GNR, GNR_CTD, GNR_1G and GNR_2G. GNR_4G and GNR_8G qualitatively provide a more accurate shape of this coastal feature. In addition, these two simulations better represent the secondary relatively denser patch on the western side. Lower density anomalies south of 39.5°N are also present in all the simulations. GNR, GNR_CTD and GNR_1G seem to qualitatively better match the reconstructed field by representing an anticyclonic eddy around a local density minimum, with an approximate 40 km diameter. However, the exact shape of this
anomaly and in particular its meridional extension, was not properly observed during Leg 3, which only provided a single Scanfish zonal section at 39.4°N across this anomaly. While it is represented as a close eddy in the reconstructed field due to the interpolation method, the more elongated shape in the meridional direction provided by GNR_4G and GNR_8G is also consistent with the Scanfish observations. Notice that the data-assimilative simulations all qualitatively improve the solution without DA. Among them, GNR_4G and GNR_8G provide a particularly remarkable pattern agreement with the Scanfish and
CTD observations.

To quantify the improvement, we now present the normalized RMSD, both considering the type of large scale observations which were assimilated over the whole domain, and the independent sea trial observations used in this section. We computed the normalized RMSD for each of the seven data-assimilative simulations on 22 June and for the different sources of observation (Figure 12). CTD and Scanfish observations between 20 and 23 June were considered synoptic for this purpose.

As already described in Section 3.1, the generic assimilation (SLA along-track, SST and Argo) provides similar results over the whole domain to that obtained during the spinup period, when no dense profile data is assimilated in the REP14-MED domain. It reduces significantly the RMSD compared to the NO_ASSIM simulation. Moreover, it also allows to reduce by around 10% the RMSD against independent CTD observations both in temperature and salinity. While it improves the comparison with Scanfish temperature observations, it slightly degrades salinity comparisons.

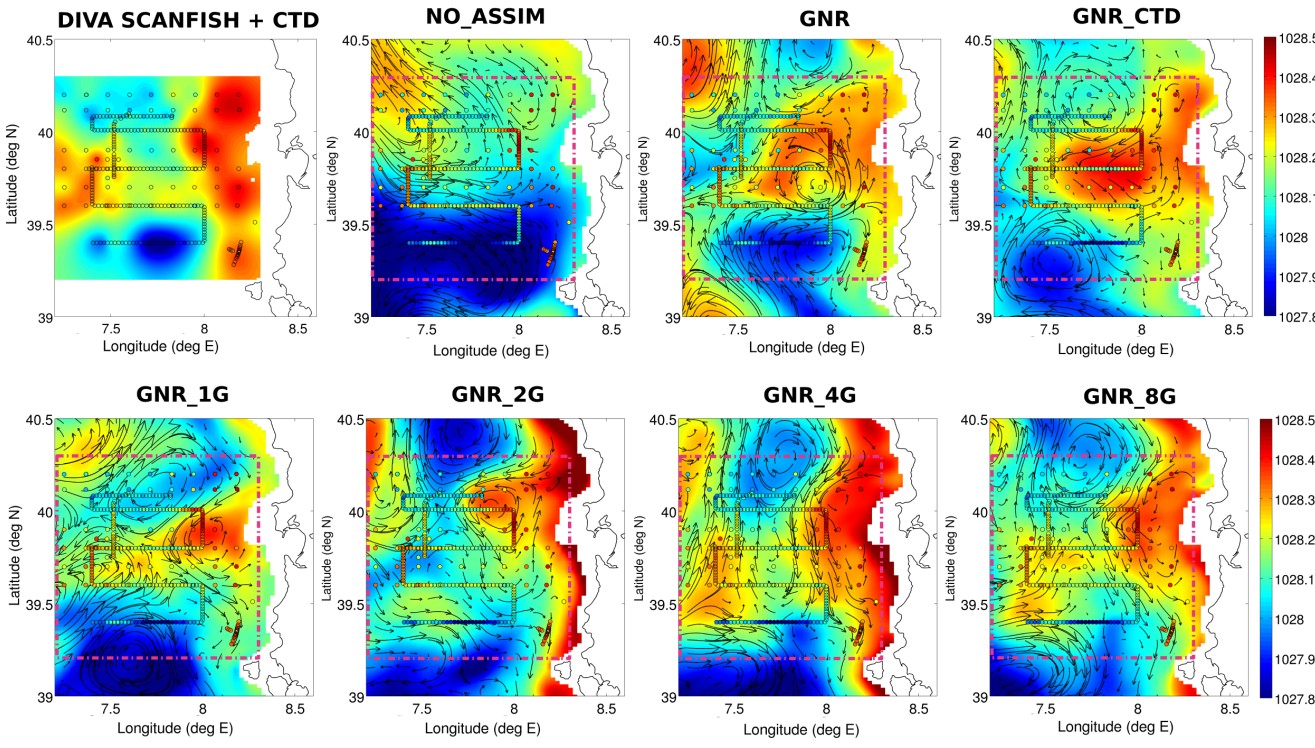

**Figure 11.** Top left panel: potential density field reconstruction from Scanfish and CTD data collected between 20 and 23 June (kg $\cdot$ m$^{-3}$). Remaining panels: potential density (kg $\cdot$ m$^{-3}$) and model currents at 50m depth on 22 June for the seven simulations NO_ASSIM, GNR, GNR_CTD, GNR_1G, GNR_2G, GNR_4G, GNR_8G and GNR_CTD.

The ingestion of high-resolution local data from the REP14-MED campaign further reduces the RMSD with SST, SLA and Argo computed over the whole domain, with similar results when assimilating observations from CTDs and gliders (with the exception of the SLA which is not improved when a single glider is assimilated). In the REP14-MED domain, the assimilation of CTDs allows a reduction of the RMSD against independent observations between 30 and 40%, both in temperature and
5  salinity, with respect to the simulation without any DA. The assimilation of glider observations also reduces the RMSD, with an overall enhanced performance as the number of platforms increases. The comparison with different platforms and variables provides slightly different rankings of the simulations. For instance, in this comparison, GNR_1G provides a similar performance as GNR_CTD against independent CTD temperature data, but a lower performance against CTD salinity and Scanfish measurements. The assimilation of data from four gliders improves the performance with respect to the assimilation of
10  CTDs when comparing to Scanfish salinity data, but the performance is lower when comparing to the other sources. GNR_CTD provides the best RMSD reduction when comparing to CTD salinity and Scanfish temperature, but it is GNR_8G which shows

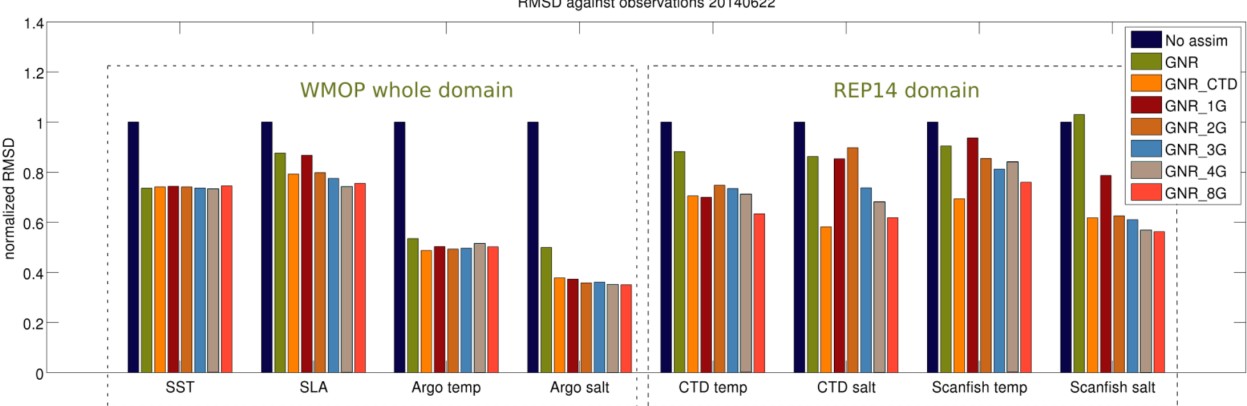

**Figure 12.** Normalized RMSD against observations on 22 June for the seven numerical simulations. Dashed bounding boxes delimitate on the one side the observations assimilated over the whole domain, and on the other side the independent campaign observations within the REP14-MED domain.

the best performance when considering CTD temperature and Scanfish salinity data. These variations are probably due to the specific spatial sampling of the CTDs and Scanfish (see Figure 2) combined to the high spatial oceanic variability in the area.

An average RMSD reduction number is obtained here by computing the square root of the average normalized mean square difference over the four comparisons (CTD and Scanfish temperature and salinity) in the REP14-MED domain. These synthetic average normalized RMSD scores are presented in Figure 13. This average RMSD gives scores of 39% and 40% of error reduction for the simulations GNR_CTD and GNR_8G, respectively. According to this overall metric, the CTD survey is more performant than a sampling using four gliders, and shows very close performance as that obtained with eight gliders. Notice that the average normalized RMSD illustrates the progressive gain obtained when increasing the number of gliders considered in these experiments.

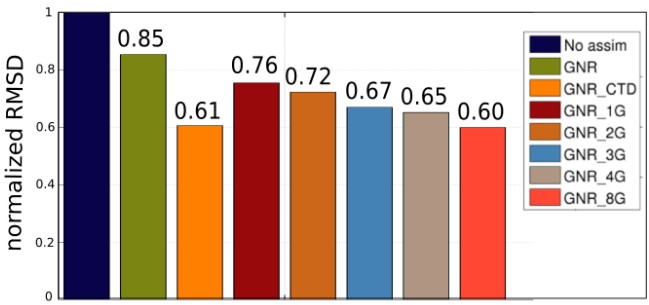

**Figure 13.** Average normalized RMSD against independent observations in the REP14-MED areas on 22 June for the seven numerical simulations.

## 4 Discussion

The assimilation of observations is crucial to improve forecasts. Regional forecasting systems should be able to efficiently combine high-resolution local profile data and larger scale satellite observations over an extended modelling domain. The recursive Ensemble Optimal Interpolation scheme employed in this study is shown to be able to ingest both types of data and to systematically reduce the errors when compared to a control free-run simulation. Even if the EnOI is theoretically inferior to more advanced DA schemes such as the ensemble Kalman filter or the 4DVar, its numerical efficiency makes it a good compromise for operational and practical implementations with high-resolution models.

The domain localization approach, which does not take into account the observations further than a given radius to correct the field at a given location, guarantees that the assimilation of dense profile observations from gliders and CTDs over a reduced area does not degrade the results over the whole modelling domain. Moreover, it allows to reduce the RMSD and to improve the representation of local water masses and the associated circulation in the reduced REP14-MED area which has dimensions around 100 km. The corrections introduced by the assimilation of CTD data during Leg 1 are found to remain in time, providing a very positive and significant error reduction when comparing to independent measurements 10 days after the initial CTD data collection.

As a limitation, we notice that the oceanographic structures of small horizontal and vertical dimensions, which have a strong signature in the dense observation datasets, are only approximately represented in the temperature and salinity fields just after assimilation, as shown in Figure 10). This is the case for instance for the strong positive temperature anomaly around 7.7°E-39.3°N, which is the signature of an eddy with an horizontal diameter around 40 km and a vertical dimension around 50 m. We attribute this limitation on the one hand to the smoothing effect of the background error covariances, which impacts both along the horizontal and vertical directions, and on the other hand to the nudging initialization procedure, which attenuates the model correction with the aim to provide more dynamically consistent fields. To illustrate the error covariances of our EnOI implementation, Figure 14 shows both the horizontal and vertical model ensemble correlations generated from the ensemble for the analysis on 22 June.

A spatial smoothing takes place during the assimilation, affecting the area with significant correlations with the observed locations. The ensemble correlation distances are found to be around 100 m in the vertical and 75 km in the horizontal, being then larger than the smaller scale structures observed in the CTD and glider surveys. Two-steps assimilation strategies separating long- and short-distance correlation scales might allow to improve the representation of these finer patterns in more sophisticated DA systems (e.g. Li et al. (2015)).

The second factor explaining this limitation is related to the nudging initialization strategy, which has the advantage of limiting undesired model shocks after the analysis, but also attenuates the corrections and therefore the agreement with observations. The simple nudging technique used in this work is easy to implement and cost-effective. It could be improved in the future by considering more advanced approaches (e.g. Sandery et al. (2011)).

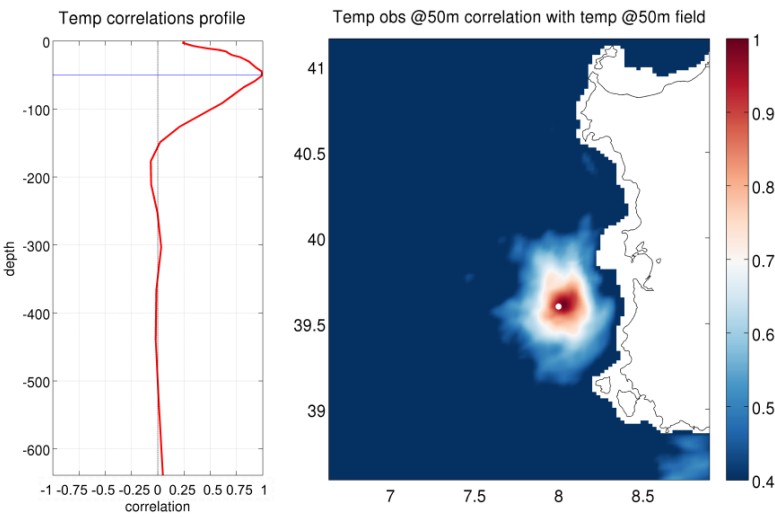

**Figure 14.** EnOI temperature ensemble correlations for a temperature observation within the REP14-MED domain at 50 m depth (position indicated by the white dot on the right panel). Left: correlations along the vertical. Right: horizontal correlations at 50m depth.

In spite of these limitations, the EnOI scheme implemeted in this study is shown to be able to properly ingest the multi-scale observations, which leads to improved representations of the mesoscale structures in the REP14-MED area and enhanced forecasting skills persisting several cycles after the assimilation of the dense observations.

While CTDs allow a relatively fast comprehensive description of a specific study area, gliders provide a slower sampling but also allow a repetition of specific monitoring tracks over a longer period. In this study, the CTD initialization survey results in a similar forecast performance after DA in terms of RMSD reduction as an 8-glider continuous monitoring of the area flying along predefined paths with regular spacing. It should be highlighted that the meridional spacing in the case of the 8 gliders fleet is the same as for the CTD casts ( 10 km). The improvement provided by the higher spatial resolution offered by gliders in the zonal direction might be limited by the spatial resolution of the model, which does not allow to ingest the very fine-scale features observed by the gliders. In that sense, it is likely that glider DA would further benefit from an increase of the model resolution. It should be mentioned that while glider platforms are considered autonomous, their operation still implies a very significant effort in terms of platform deployment, recovery, piloting and maintenance. The models could also highly benefit from the near real-time controllability of gliders, allowing to continuously adjust their path along optimal routes in the study area. In this framework, efficient adaptive sampling procedures should theoretically allow to use a reduced number of gliders to reach the same level of performance (Lermusiaux, 2007) and lead to a better description of specific targeted features, as long as their representation is permitted by the model resolution. The definition of optimal collective behaviours based for instance on glider fleet coordination or cooperation (e.g. Alvarez and Mourre (2014)) also constitutes an interesting field of research in that direction.

## 5   Conclusions

We presented in this work the results of several simulations assimilating different multi-platform observations in the context of the REP14-MED sea trial carried out in June 2014 off the West coast of Sardinia. The experiments were designed to assimilate intensive campaign data from CTDs and gliders, along with satellite SST and SLA, as well as Argo profile observations over the whole model domain covering the Western Mediterranean Sea from Gibraltar to the Sardinia Channel. The objective was to explore the performance of different sampling strategies based on either a dense CTD initialization survey or a glider fleet sampling, in improving model forecasting capabilities in a specific area.

The DA system was shown to perform correctly. The Local Multimodel EnOI scheme, following three day recursive cycles with a one day nudging initialization phase after analysis, allows to properly ingest both large scale data over the whole Western Mediterranean domain and high density temperature and salinity profiles collected during the sampling experiment over a limited area. In spite of the limitations associated with the smoothing effect of ensemble covariances, which do not allow to exactly represent the smaller scale features present in the observations, this system enables a significant improvement of the forecasting skill of the model with respect to the simulation without assimilation, and that assimilating only satellite and Argo data. Its reduced cost makes it a good option for operational implementations.

While the assimilation of generic observations from SST, SLA and Argo leads to an average error reduction of 15% when comparing to independent measurements collected during Leg 3 of the sea trial in the REP14-MED area, the assimilation of glider and CTD data allows an additional significant improvement. Gliders, which provide a continuous sampling of the area along regularly spaced zonal tracks, allow to reduce the forecast error as the number of platforms increases. The consideration of one glider leads to a 24% average error reduction with respect to the simulation without assimilation. This percentage increases to 28%, 33%, 35% and finally 40% for the two-glider, three-glider, four-glider and eight-glider fleet configurations, respectively. Incrementing the number of gliders results in a better representation of the ocean state captured by observations, with a most accurate representation of the mesoscale structures and associated circulation.

The assimilation of the observations from the dense initialization survey based on 10-km spaced CTD stations leads to an average error reduction of 39%: it outperforms the four-glider configuration and provides very similar results in terms of RMSD as the eight-glider fleet configuration. The 10 km spacing offered by both sampling strategies is essential here to improve the representation of the mesoscale variability in the study area. In view of these results, gliders certainly provide a very interesting alternative to traditional CTD surveys used to initialize high-resolution regional ocean models, provided that a fleet of vehicles can be deployed at sea. Moreover, an increased performance can certainly still be expected by optimizing the regular track sampling carried out in this experiment through adaptive sampling procedures.

*Acknowledgements.*   The authors especially thank Reiner Onken for leading the experiment. They also acknowledge all the partners, scientists and technicians having participated to the REP14-MED sea trial, allowing the collection and processing of this very valuable dataset. They especially thank Ines Borrione and Michaela Knoll for their help in providing and interpreting ADCP data. This work uses CMEMS Products from the Mediterranean Monitoring and Forecasting Centre (MED-MFC) which produces the Mediterranean Forecasting System. The Hirlam

atmospheric fields used to force the ocean forecasts are provided by the Spanish National Meteorological Agency. SOCIB DA system developments were partly supported by the Medclic project funded by La Caixa foundation and the JERICO-NEXT European project.

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
