# Peer review of "Dense CTD survey versus glider fleet sampling: comparing data assimilation performance in a regional ocean model West of Sardinia"

_Ocean Science, 2018_

## Referee Comment (RC1) · Anonymous Referee #1 · 11 May 2018

General Comments: The manuscript investigates the impact of the assimilation of different datasets on simulations and forecasts performed with a mesoscale resolving regional model (ROMS) implemented in the Western Mediterranean sea. I generally like the paper. Authors investigate the impact of new technologies and sampling strategies on the description of the ocean state. My main concern is about the behaviour of the model without assimilation. In particular, as the authors state (p.12 l.2-4) "min and max are shifted by 0.2 kg/m3 in the panels corresponding to the simulations due to a persistent bias in the model density field": it seems that something is going wrong with

temperatures. Temperatures at 50 m (Top-left panel of fig.8) seem up to 2 degree celsius warmer than data, that is quite a lot for this depth and period. I wonder if this could be an issue related to surface fluxes and/or to a bad vertical discretization (maybe the model fails to match stratification and thermocline position). Or, it could be a BIAS due to the lenght of the (drifting) simulation? I ask the Authors AT LEAST to discuss the sources of such bias and possible (future or present) ways to fix it. Further, to better understand the relevance of such bias on simulations (and thus the impact of assimilation) it would be good to see also layers shallower than 50 m: I suspect an even higher temperature and density bias on shallower layers. If the BIAS is larger than 1-2 °C it should be better to fix it BEFORE assimilation.

Specific Comments, grammar and typos: Title: I would suggest a change in the title as in its present form it misses to inform about Analyses/Data assmilation. Something like "Sampling strategy does matter: impact of assimilation of glider data compared to dense CTD survey in a regional ocean model West of Sardinia". Or maybe shorter but I suggest with the information of the performed ocean synhteses.

Abstract is ok.

Intro and or Methods: The paper misses a description of the known circulation of the area. There are papers specifically devoted to the circulation of the Sardinian sea (Ribotti et al. 2004 about surface mesoscale circ; Testor et al. 2003, 2005 about LIW transport mediated by Sardinian eddies; Olita et al. 2013 about surface circulation and upwelling; papers already published in the same special issue as the present manuscript; some other references to the area can be found in Mediterranean-scale studies). -p4 line 17: a description/reference of the EnOI method and algorithms is missing. Is the software developed by Authors? If yes, it should be referenced or described (even in an Appendix)- -p6 line 30: "an horizontal" should be "a horizontal" -p8 line 5: it is not clear if the observations used for RMSD are independent (i.e. are them also assimilated during spinup?). -P15 line 17: "The domain localization approach guarantees that the assimilation of dense profile observations from gliders and

CTDs over a reduced area does not degrade the results over the whole modelling domain". Please repeat also here what the "domain localization" procedure/alghorithm do. PLease also show somewhere the impact of such procedure on the whole modelling domain (for instance by showing some sensitivity test to the chosen radius) -p17 line 7-9 " In this study, the CTD initialization survey results in a similar forecast performance after data assimilation as an 8-glider continuous monitoring of the area flying along predefined paths with regular spacing": Please also specify that such "regular spacing" is the same as the meridional spacing of CTD sampling grid. This is important to be specified as it seems that the larger number of data (higher along trasect resolution) from glider data does not imply an improved ocean state description after assimilation, which on the contrary is "just" function of the maximum resolution of your grid (resolution is larger across transect than along transect). This would suggest of course that CTD sampling would benefit by an equally spaced grid, and that probably glider data assimilation would benefit a finer maximum resolution (that could be able to catch submesoscale you observed to be smoothed by your system). -p18: Adaptive sampling procedure (finalized to assimilation) would probably lead to better description of some specific features BUT with the limitations of what said here above, i.e. limited (in resolving features) by the maximum resolution of the sampling grid in a given area (combined with the resolution of the model itself).

---

## Short Comment (SC1) · 16 May 2018

Could you please correct the citation of the CMEMS Mediterranean Sea Reanalysis? The year is 2014 and the doi is wrong. Here the correct citation.

Simoncelli, S., Fratianni, C., Pinardi, N., Grandi, A., Drudi, M., Oddo, P., & Dobricic, S. (2014). "Mediterranean Sea physical reanalysis (MEDREA 1987-2015)". Data set. E.U. Copernicus Marine Service Information. DOI: https://doi.org/10.25423/medsea_reanalysis_phys_006_004

[Figure]

Thank you, Simona Simoncelli

---

## Referee Comment (RC2) · Anonymous Referee #2 · 27 Jun 2018

General Comments:

In this work, the authors present results from different simulations assimilating data of several observational arrays in synergy with glider data. The analysis includes a high-resolution regional model for the Western Mediterranean, with a focus in the coastal area West of Sardinia. The system incorporates a Local Multi-Model Ensemble Optimal Interpolation scheme to ingest satellite and dense in-situ data. The concluding remark is that an optimized sampling strategy of a gliders fleet in the future can significantly increase the Data Assimilation (DA) performance of an operational application.

[Figure]

The manuscript is clear, concise and well written. The study shows some interesting results and supports well the argument of designing glider missions in synergy with other observational platforms to increase DA performance. However, I have one main concern regarding the performance of the stand-alone ocean model without DA, which in turns raises some questions for the DA post-analysis correction. Overall, I find the manuscript worthy of publication, after a major revision. Please find below a list of comments that I would like the authors to address. My first two specific comments are the most important ones.

Specific comments:

1) My main concern is the performance of the regional ocean model WMOP without DA. The authors provide a schematic of the main circulation features for the whole WMOP domain (i.e. Fig. 2 top panel), but model performance and outputs are only presented for the coastal area REP14-MED (i.e. Figs. 2 bottom panels, 4, 8, 9, 12). This is not a good practice, especially when the free run appears not to represent adequately the coastal dynamics of the REP14-MED domain (there are large biases, e.g. page 12 lines 2-3, and completely different circulation patterns before and after DA). The authors should at least provide a validation section of the regional model over the whole WMOP domain without DA. In my view, it is acceptable to have a well-tuned regional model (e.g. like WMOP), even if it fails in some coastal areas (e.g. in REP14-MED domain).

2) Following my first comment, the implications of having a biased model coupled with a DA system can be significant. For instance, a DA platform usually incorporates a convex scheme, and therefore it will always return an analysis correction. The main question is if this correction actually has a physical meaning (even if the RMSD error is reduced after DA, as it is the case in this study). Especially, in ensemble-based DA systems one should show that model and data pdfs overlap (at least partially). I would like the authors to illustrate that the model ensemble spread has joint probabilities with the assimilated observations, taking under account their errors mentioned in the text

(e.g. page 6 line 32). This could be done providing innovation/misfit statistics in data space for some variables (e.g. at least one from SST, SLA, T, S), over a period and an area of the authors preference (e.g. an area covered from satellite observations and/or glider/Argo profiles).

3) The title should reflect the fact that the study focuses on data assimilation, e.g. "... comparison of data assimilation performance..." or something like that.

4) The "section 2.2." is clear, but quite compact when discussing the DA scheme. In a DA paper, it is always useful to present one or two equations (not more) of the analysis kernel, since there are several sub-optimal variants of the EnKF (e.g. SEEK, LETKF, EnOI, SEIK etc.).

5) In "section 2.2." the initial state and ocean boundary conditions of the WMOP are discussed (page 4 lines 12-14). The use of the CMEMS-MED reanalysis is an appropriate option to provide initial/boundary conditions for the seven-year long free run hindcast simulation (used later on to calculate BECs). However, for the seven sensitivity DA experiments, spanning the short period 1-24 June 2014, the analysis CMEMS-MED perhaps would have been a better option (perhaps also the biases would have been smaller). I would like the authors to justify their choices in terms of initial/boundary conditions for the DA short simulations.

6) Page 6 line 1 "80-member ensemble". Calculating BECs by sampling long simulations it's a nice not expensive alternative compared to stochastic flow-dependent ensembles, but the degrees of freedom are eventually lesser than actually having an ensemble of 80-members (like in an EnKF system for instance). I think the most appropriate terminology in this case is "modes" or "realizations" instead of "members".

7) Page 6 line 26 "original 1-km resolution data is smoothed and interpolated onto 10km-resolution grid to limit the number of observations". This a common strategy in most DA systems to reduce the computational cost in each assimilation cycle. The most common options are "super-obbing" i.e. averaging/smoothing data like in this

study or "thinning", i.e. just sub-sampling the data. Can the authors justify why they choose the one over the other option?

8) Page 10 line 1 "does not negatively affect the overall performance of the system". The authors are correct with the word "not negative", since in both cases (with/without T, S assimilation) the RMSD compared to the free run is reduced for all variables. But, DA performance for SLA clearly reduces when T ,S are assimilated (see Figs 6 and 7 for SLA). Perhaps, this is not something surprising (covariances can be contaminated for SLA when T, S are injected), but I would like the authors to discuss this effect.

9) Page 11 lines 7-8 "GNR simulation redistributes these water masses over the domain". In my view, the NO_ASSIM and GNR circulation patterns and water masses are completely different in the REP14-MED domain. I don't see it as a redistribution of these specific water masses. Is this a local coastal effect over the REP14-MED or perhaps a remote effect over the whole WMOP region? Please clarify in the text.

Best regards.

---

## Author Comment (AC1) · 7 Aug 2018

Thanks. The citation has been corrected
* * *

---

## Author Comment (AC2) · 7 Aug 2018

**Reviewer:** General Comments: The manuscript investigates the impact of the assimilation of dif- ferent datasets on simulations and forecasts performed with a mesoscale resolving regional model (ROMS) implemented in the Western Mediterranean sea. I generally like the paper. Authors investigate the impact of new technologies and sampling strate- gies on the description of the ocean state. My main concern is about the behaviour of the model without assimilation. In particular, as the authors state (p.12 l.2-4) "min and max are shifted by 0.2 kg/m3 in the panels corresponding to the simu-

lations due to a persistent bias in the model density field": it seems that something is going wrong with temperatures. Temperatures at 50 m (Top-left panel of fig.8) seem up to 2 degree cel- sius warmer than data, that is quite a lot for this depth and period. I wonder if this could be an issue related to surface fluxes and/or to a bad vertical discretization (maybe the model fails to match stratification and thermocline position). Or, it could be a BIAS due to the lenght of the (drifting) simulation? I ask the Authors AT LEAST to discuss the sources of such bias and possible (future or present) ways to fix it. Further, to better understand the relevance of such bias on simulations (and thus the impact of assimila- tion) it would be good to see also layers shallower than 50 m: I suspect an even higher temperature and density bias on shallower layers. If the BIAS is larger than 1-2$\deg$ C it should be better to fix it BEFORE assimilation.

**Response:**

We first would like to thank the reviewer for her/his positive feedback and comments which we think have helped us to improve the manuscript.

We fully agree that model bias is a major concern for this kind of data assimilation study and so completed the paper with a better description of this particular aspect in our system.

In particular, the 0.2kg/m3 difference found between the simulations and the density field inferred from the observations for the period 20-23 June (Fig. 9) remained unexplained in the initial version of the manuscript. After a careful revision of innovations, model and observed temperature and salinity fields, we could not find the theoretically corresponding differences in temperature and salinity. Looking into further details, we identified the source of this bias as an error in the computation of the potential density, related to a different reference level considered for the model and observations. This error, which was mainly affecting the range of colorbars in Fig. 9 (now Fig. 11 in the revised version), has been fixed in the revised manuscript. Once this correction applied, there is no remaining density bias between the model results after data assimilation

and Scanfish and CTD observations. Density values on CTD stations collected on 20 June north of the Scanfish tracks were also added to the DIVA reconstruction, allowing to extend the coastal fringe of relatively denser water. Notice that these stations were already considered in the initial version of manuscript for the computation of the RMSD.

As mentioned by the reviewer, the second important point concerned the apparent temperature bias in the model without data assimilation. To this respect, we have added in the manuscript pictures of the SST from model and observations over the whole domain at the beginning of the simulation, as well as a map of the differences, and histograms of SST, SLA, T and S innovations. Positive and negative SST differences are found all over the domain, illustrating that no significant bias affects the data assimilation system, which is applied over the whole domain. When focusing on the REP14 area, local positive differences are found (the model being warmer than the observations, also represented by negative innovations), but these are related to local processes and far from systematic over the whole domain. The pdf of innovations (figure 5) also show that no significant bias is affecting the analysis.

We have also changed the figure presenting the general circulation in the domain so as to illustrate the model general circulation (Figure 1). The free run model has been deeply evaluated in other studies which are in the process of peer-reviewed publication.

**Reviewer:** Specific Comments, grammar and typos: Title: I would suggest a change in the title as in its present form it misses to inform about Analyses/Data assmilation. Something like "Sampling strategy does matter: impact of assimilation of glider data compared to dense CTD survey in a regional ocean model West of Sardinia". Or maybe shorter but I suggest with the information of the performed ocean synhteses.

**Response:** We have changed the title.

**Reviewer:**

Abstract is ok. Intro and or Methods: The paper misses a description of the known circulation of the area. There are papers specifically devoted to the circulation of the Sardinian sea (Ribotti et al. 2004 about surface mesoscale circ; Testor et al. 2003, 2005 about LIW transport mediated by Sardinian eddies; Olita et al. 2013 about surface circula- tion and upwelling; papers already published in the same special issue as the present manuscript; some other references to the area can be found in Mediterranean-scale studies).

**Response:**

We have extended the description of the circulation in the area and added some of the suggested references.

**Reviewer:**

-p4 line 17: a description/reference of the EnOI method and algorithms is missing. Is the software developed by Authors? If yes, it should be referenced or de- scribed (even in an Appendix)-

**Response:**

The description of the data assimilation system has been enlarged and been separated in a new section "2.3 – Data Assimilation system" . References of former applications of the code which was adapted for the present study have also been added.

**Reviewer:**

-p6 line 30: "an horizontal" should be "a horizontal"

**Response:**

Done.

**Reviewer:**

-p8 line 5: it is not clear if the observations used for RMSD are independent (i.e. are them also assimilated during spinup?).

**Response**:

The "generic" observations (SLA, SST, Argo) were also assimilated during the spinup period. This point is highlighted in page 9 l.11-12 "A spinup period of 9 days was imposed for all these data-assimilative simulations, during which only the generic observations were assimilated" , and illustrated in figure 7, were the timeline of the spinup and data-assimilative simulations is represented.

**Reviewer:**

-P15 line 17: "The domain localization approach guarantees that the assimilation of dense profile observations from gliders and CTDs over a reduced area does not degrade the results over the whole modelling domain". Please repeat also here what the "domain localization" procedure/alghorithm do. PLease also show somewhere the impact of such procedure on the whole modelling domain (for instance by showing some sensitivity test to the chosen radius)

**Response:**

We have extended in Section 2.3 the explanation of the domain localization approach used in this study. We also add here some pictures from sensitivity tests that were performed to illustrate the effects of the localization. Figure 1, from this review, shows the temperature fields at 50m depth of two experiments assimilating CTDs with 200 and 40km of localization radius respectively. The proper description of these effects would require a dedicated study. We think that this aspect remains out of the main scope of the paper and decided not to include these figures in the manuscript.

**Reviewer:**

-p17 line 7-9 " In this study, the CTD initialization survey results in a similar forecast performance after data assimilation as an 8-glider continuous monitoring of the area flying along predefined paths with regular spacing": Please also specify that such "regular spacing" is the same as the meridional spacing of CTD sampling grid. This is important to be specified as it seems that the larger number of data (higher along trasect resolution) from glider data does not imply an improved ocean state description after assimilation, which on the contrary is "just" function of the maximum resolution of your grid (resolution is larger across transect than along transect). This would suggest of course that CTD sampling would benefit by an equally spaced grid, and that probably glider data assimilation would benefit a finer maximum resolution (that could be able to catch submesoscale you observed to be smoothed by your system).

**Response:**

This is a very interesting point. A sentence has been added in the discussion: "It should be highlighted that the meridional spacing in the case of the 8 gliders fleet is the same as for the CTD casts ( 10km). The improvement provided by the higher spatial resolution offered by gliders in the zonal direction might be limited by the spatial resolution of the model, which do not allow to ingest the very fine-scale features observed by the gliders. In that sense, it is likely that glider data assimilation would further benefit from an increase of the model resolution."

**Reviewer:**

-p18: Adaptive sampling procedure (finalized to assimilation) would probably lead to better description of some specific features BUT with the limitations of what said here above, i.e. limited (in resolving features) by the maximum resolution of the sampling grid in a given area (combined with the resolution of the model itself).

**Response:**

These limitations have been added in the text.
* * *
[Figure]

[Figure]

**Fig. 1.** Temperature fields at 50m depth for two different experiments assimilating CTDs with a localization radius of 40km (left pannel) and 200km (rigth pannel)

---

## Author Comment (AC3) · 7 Aug 2018

**Reviewer:**

General Comments:

In this work, the authors present results from different simulations assimilating data of several observational arrays in synergy with glider data. The analysis includes a high-resolution regional model for the Western Mediterranean, with a focus in the coastal area West of Sardinia. The system incorporates a Local Multi-Model Ensemble Optimal

[Figure]

Interpolation scheme to ingest satellite and dense in-situ data. The concluding remark is that an optimized sampling strategy of a gliders fleet in the future can significantly increase the Data Assimilation (DA) performance of an operational application.

The manuscript is clear, concise and well written. The study shows some interesting results and supports well the argument of designing glider missions in synergy with other observational platforms to increase DA performance. However, I have one main concern regarding the performance of the stand-alone ocean model without DA, which in turns raises some questions for the DA post-analysis correction. Overall, I find the manuscript worthy of publication, after a major revision. Please find below a list of comments that I would like the authors to address. My first two specific comments are the most important ones.

**Response:**

We acknowledge the reviewer for her/his constructive comments which have helped us to improve the manuscript.

**Reviewer:**

Specific comments:

1) My main concern is the performance of the regional ocean model WMOP without DA. The authors provide a schematic of the main circulation features for the whole WMOP domain (i.e. Fig. 2 top panel), but model performance and outputs are only presented for the coastal area REP14-MED (i.e. Figs. 2 bottom panels, 4, 8, 9, 12). This is not a good practice, especially when the free run appears not to represent adequately the coastal dynamics of the REP14-MED domain (there are large biases, e.g. page 12 lines 2-3, and completely different circulation patterns before and after DA). The authors should at least provide a validation section of the regional model over the whole WMOP domain without DA. In my view, it is acceptable to have a welltuned regional model (e.g. like WMOP), even if it fails in some coastal areas (e.g. in REP14-MED domain).

**Response:**

This is linked to our answer to reviewer#1. We have added a new section aiming at providing elements of validation of the model over the whole domain, in particular illustrating the absence of model bias at the basin-scale. A more comprehensive validation of the model is out of the scope of the present paper. It is the focus of a specific work already presented in scientific congresses, and presently in the process of peer-reviewed publication.

**Reviewer:**

2) Following my first comment, the implications of having a biased model coupled with a DA system can be significant. For instance, a DA platform usually incorporates a convex scheme, and therefore it will always return an analysis correction. The main question is if this correction actually has a physical meaning (even if the RMSD error is reduced after DA, as it is the case in this study). Especially, in ensemble-based DA systems one should show that model and data pdfs overlap (at least partially). I would like the authors to illustrate that the model ensemble spread has joint probabilities with the assimilated observations, taking under account their errors mentioned in the text (e.g. page 6 line 32). This could be done providing innovation/misfit statistics in data space for some variables (e.g. at least one from SST, SLA, T, S), over a period and an area of the authors preference (e.g. an area covered from satellite observations and/or glider/Argo profiles).

**Response:**

We totally agree with the concern of the reviewer. To better clarify background model errors, we have added a figure showing the innovations in terms of SST, SLA, T and

[Figure]

S for the first analysis of the spinup period (figure 4 in the article). This shows that 1) there is no significant bias over the whole domain, and 2) the magnitude of the innovations is in agreement with the prescribed observation errors and ensemble spread. Thus, we are confident that the system is properly calibrated at the scale of the model domain over which the analysis is performed. The apparent bias in the REP14 area is due to local (in space and time) differences associated with the regional dynamics. This is precisely what we expect the data assimilation system to be able to correct, as it applies over a larger domain where these errors compensate.

**Reviewer:**

3) The title should reflect the fact that the study focuses on data assimilation, e.g. "... comparison of data assimilation performance..." or something like that.

**Response:**

We have changed the title.

**Reviewer:**

4) The "section 2.2." is clear, but quite compact when discussing the DA scheme. In a DA paper, it is always useful to present one or two equations (not more) of the analysis kernel, since there are several sub-optimal variants of the EnKF (e.g. SEEK, LETKF, EnOI, SEIK etc.).

**Response:**

The description of the model has been separated in a new section "2.3 – Data Assimilation system" and some equations have been introduced to clarify it.

**Reviewer:**

5) In "section 2.2." the initial state and ocean boundary conditions of the WMOP are discussed (page 4 lines 12-14). The use of the CMEMS-MED reanalysis is an appropriate option to provide initial/boundary conditions for the seven-year long free run hindcast simulation (used later on to calculate BECs). However, for the seven sensitivity DA experiments, spanning the short period 1-24 June 2014, the analysis CMEMS-MED perhaps would have been a better option (perhaps also the biases would have been smaller). I would like the authors to justify their choices in terms of initial/boundary conditions for the DA short simulations.

**Response:**

Some hindcast sensitivity tests have been performed using both analysis and reanalysis fields for initial and boundary conditions. A major difference is the consideration of atmospheric pressure forcing in CMEMS-MED analysis, and not in the re-analysis. This introduced some unrealistic high-frequency signal in terms of SLA from the open boundaries of the WMOP domain when using the analysis. This is the reason why we worked in this study with CMEMS-MED reanalysis fields.

**Reviewer:**

6) Page 6 line 1 "80-member ensemble". Calculating BECs by sampling long simulations it's a nice not expensive alternative compared to stochastic flow-dependent ensembles, but the degrees of freedom are eventually lesser than actually having an ensemble of 80-members (like in an EnKF system for instance). I think the most appropriate terminology in this case is "modes" or "realizations" instead of "members".

**Response:**

The word "members" has been substituted by "realizations"

**Reviewer:**

7) Page 6 line 26 "original 1-km resolution data is smoothed and interpolated onto 10km-resolution grid to limit the number of observations". This a common strategy in most DA systems to reduce the computational cost in each assimilation cycle. The most common options are "super-obbing" i.e. averaging/smoothing data like in this study or "thinning", i.e. just sub-sampling the data. Can the authors justify why they choose the one over the other option?

**Response:**

We generally consider super-obbing as a more appropriate approach since it theoretically allows to reduce the uncorrelated observation errors before assimilation. This is why we applied this approach in this study. However, we didn't perform any sensitivity test to this particular aspect.

**Reviewer:**

8) Page 10 line 1 "does not negatively affect the overall performance of the system". The authors are correct with the word "not negative", since in both cases (with/without T, S assimilation) the RMSD compared to the free run is reduced for all variables. But, DA performance for SLA clearly reduces when T ,S are assimilated (see Figs 6 and 7 for SLA). Perhaps, this is not something surprising (covariances can be contaminated for SLA when T, S are injected), but I would like the authors to discuss this effect.

**Response:**

Page 12 l.15-17: "Notice that the relatively larger SLA RMSD found during the period 10-23 June compared to the spinup period also affects the GNR simulation. Therefore, it is not due to the incorporation of CTD observations, but rather related to the natural evolution of SLA errors"

The relatively lower relative reduction of SLA RMSD when assimilating CTD profiles from 10 to 23 June compared to the reduction obtained during the spinup from 1

to 9 June is somehow misleading. Indeed, the same behavior is obtained when assimilating the same source of data (SST, SLA and Argo TS: GNR simulation from 10 to 23 June). This indicates that this difference is related to the evolution of the fields rather than the inclusion of high-resolution CTD profiles.

**Reviewer:**

9) Page 11 lines 7-8 "GNR simulation redistributes these water masses over the domain". In my view, the NO_ASSIM and GNR circulation patterns and water masses are completely different in the REP14-MED domain. I don't see it as a redistribution of these specific water masses. Is this a local coastal effect over the REP14-MED or perhaps a remote effect over the whole WMOP region? Please clarify in the text.

**Response:**

Altough, the apparent bias in temperature, which has already been discussed, can lead to a misunderstanding, in our opinion, the redistribution of two different water masses can de observed in the potential density fields from figures 10 and 11. Also in the salinity ones from figure 10 which, in this case, have more influence in the density structures than the temperature. This specification has been included in the text. Page 13 l.4 "As illustrated by the potential density maps..."

---

## Editor Decision (ED1)

Dear Jaime, dear Baptiste,

Please find below some remarks which might improve the technical and linguistic quality of your manuscript. For the technical corrections, see also the *Ocean Science* guidelines at https://www.ocean-science.net/for_authors/manuscript_preparation.html.

**Units**
- use negative exponents for units in the denominator (e.g. caption of Fig. 11)
- for latitude and longitude, use consistently °N or °E, not just numbers or expressions like 38.5E.
- Add a space between the value and the units, e.g. 100 km instead of 100km.
- Check also consistency of units on axes labels
- substitute 100 x 100 km by 100 km x 100 km or 100 x 100 km$^2$

**Acronyms**
- define all acronyms on first occurrence
- do not repeat definitions
- if you have defined an acronym, use the acronym throughout the manuscript

**Numbers**
- spell out consistently cardinal numbers less than 10

**Equations**
- if you refer to an equation in the text, use eq. (5) instead of equation (5)

**Figures**
- in the text body, use Figure n instead of figure n. Same for Section if you mean a special section; but: in this section, the model data are described ...

**Websites**
- if you cite web sites, add last access, e.g. http://www.socib.eu, last access 5 August 2018.

**Expressions**
- avoid too frequent usage of relative or relatively.
- use consistently Gibraltar Strait or Strait of Gibraltar, but do not mix
- there are no warm or cold temperatures; use high or low instead
- P3 L25: better with the research vessels (RV) *Alliance* and *Planet*.
- P8 L12: better assumed to be 0.0625 K$^2$ and 0.025 … (temperature differences should be expressed in Kelvin)
- P8 L12-13: remove (in practical salinity scale); this is self-evident.
- P13 L12: substitute slightly by small
- P19 L11: Let's is sloppy language

**Grammar**
- hyphenate expressions:
  high resolution → high-resolution
  fine scale → fine-scale
  long term → long-term
- do not hyphenate research vessel
- use correct past tense:
  feeded → fed
  leaded → led
- use correct plural: analysis → analyses (pl.)
- remove some capitalizations e.g. Spanish Coast, Mean Dynamic Topography, Sea Surface Temperature, Western coast, ...

**Others**
- P18 L6-7: reference to Sakov and Oke is repetitive (cf. Introduction)

- Acknowlegments: you may mention CMEMS and HIRLAM
- P2 L22-23 ← → L27: repetitive
- P2 L27 – P3 L2: you may rearrange this paragraph in a more logical manner
- Caption of Fig.1: rearrange the first sentence (bad English)
- P5 L7, 8: two times particular (style)

Cheers,
Reiner